# Verification and execution of the scientific literature via chemputation augmented by large language models

Sebastian Pagel ⓘ , Michael Jirasek ⓘ & Leroy Cronin ⓘ ✉

Chemputation is the process of programming chemical robots to do experiments using a universal symbolic language, but literature can be error-prone and hard to read due to ambiguities. Large Language Models (LLMs) have demonstrated remarkable capabilities in various domains, including natural language processing, robotic control, and more recently, chemistry. Despite significant advancements in standardizing synthetic chemistry data, automatic reproduction and verification of reported syntheses remains labor-intensive task. We introduce an LLM-based chemical research agent workflow for automatic verification of synthetic literature procedures. Our workflow can autonomously extract synthetic procedures and analytical data from extensive documents, translate these procedures into universal *X*DL code, simulate the execution in a hardware-specific setup, and ultimately execute the procedure on an *X*DL-controlled robotic system for synthetic chemistry to confirm the procedure works in the real world. This demonstrates the potential of LLM-based workflows for autonomous chemical synthesis with Chemputers. While recent LLM-based agents have demonstrated remarkable success in autonomous experiment planning, a robust workflow for the faithful digitization and verification of existing literature remains a challenge. Our approach bridges this gap by providing six realistic examples of syntheses directly executed from synthetic literature on two robotic platforms. Our workflow will significantly enhance automation in robotically driven synthetic chemistry research, streamline data extraction, improve the reproducibility, scalability, and safety of synthetic chemistry.

More than 65 million chemical reactions have been published in research papers and patents to date[1]. While being an important cornerstone, databases like Reaxys and the Open Reaction Database (ORD)[2] that attempt to capture this vast stream of data are not sufficient to tackle the rapid validation and reproduction of reported data[3]. Moreover, almost 30 million new reactions have been added to the Reaxys database since 2014 alone. Even though this might seem negligible in times when Video Generation Models and LLMs are trained on billions of examples[4,5], the necessity of manual validation of chemical reactions creates an almost insurmountable challenge. In addition to the challenges of resource limitations, time constraints, finding missing parameters, and resolving reporting ambiguities, researchers also face a vast array of specialized experimental setups and variations in terminology. There is an enormous backlog of unverified procedures, with numerous new ones being reported daily, which exacerbates the problem. Commercial and Open-Source LLMs like GPT-4, or Llama, have demonstrated impressive abilities for textual comprehension and understanding ambiguous textual data[5,6]. Beyond pure text comprehension, it was shown that these models exhibit excellent few-, one-, and zero-shot prediction abilities on unseen tasks without the need to finetune on specific tasks (in-context or gradient-free learning)[7,8]. Furthermore, a plethora of *prompting techniques* has been developed, further improving the prediction abilities of these models on various tasks[9–12]. This has led to vast amounts of research far beyond the field of Natural Language Processing into chemical research[2,13–18]. While machine-learning algorithms have contributed extensively to advances in chemistry research[19–22], the introduction of LLM as chemical agents has shown promising results in further automating chemical decision-making, experimentation, and the orchestration of self-driving laboratories[18,23–27], notably, recent work has demonstrated the power of agents to autonomously plan syntheses via web search[16] or orchestrate expert software tools to solve complex chemical tasks[18]. Especially the integration with automated chemistry platforms and human-in-the-loop

School of Chemistry, The University of Glasgow, University Avenue, Glasgow, UK. ✉e-mail: Lee.Cronin@glasgow.ac.uk

or closed-loop systems, offers a transformative pathway to accelerate experimental validation and exploration[28–39].

Herein, we present Autonomous Chemputer Reaction Agents (ACRA), a LLM-based chemical multi-agent workflow, automating the tedious process of extracting reaction data from literature procedures, standardizing them, translating them into robotic instructions, conducting experiments, and iterative *XDL* code development suggestions (Fig. 1).

ACRA is set up to translate synthetic procedures into autonomously executable procedures via the Chemical Description Language (*XDL*), which represents synthetic steps, reagents, as well as available hardware in an unambiguous way (compare Fig. 2C for a simplified *XDL* which is directly mapped to an abstract hardware configuration)[40]. While early agents[16,18] have excelled at de novo experiment planning and direct code generation for specific platforms, a critical challenge remains in the faithful digitization of the existing literature. Unlike prior efforts that focus on generating new routes or utilizing platform-specific APIs, our workflow is designed to standardize the vast corpus of unstructured reported procedures. ACRA enables the transition all the way from a static literature document to the execution of a synthetic procedure by translating into the hardware-agnostic *XDL* standard, and importantly, introduces a simulation layer to validate that the generated instructions match physical hardware constraints. In previous work, it was shown how synthetic procedures represented in *XDL* can be used to unambiguously validate synthetic procedures[41]. Similar to prior work[42], our workflow implements a *paper-scraping-agent* extracting synthesis procedures alongside purification information, analytical information, and further related information by iteratively analysing parts of a literature text and creating a knowledge-graph (KG) of the given source. The *procedure-agent* subsequently sanitizes extracted procedures by identifying potential ambiguities and provides additional physicochemical information from chemical databases. Additionally, all procedures are classified in one of three categories (executable, reaction blueprint, or incomplete procedure) to identify procedures that can be automatically executed without further intervention. As executable or blueprint-identified procedures are then translated into *XDL* by the *XDL-agent* and iteratively corrected with feedback from a three-stage validation workflow. Finally, successfully translated procedures can be executed

on a Chemputer platform[29,43,44]. Synthetically verified procedures are stored in an *XDL* database, systematically increasing the number of validated and standardized procedures, greatly lowering the barrier of reproduction for other chemists. The procedures and their associated *XDL* are used to guide future translation by acting as examples for the *XDL-agent*, which implements a few-shot prompting style. All generated and extracted data are stored in a unified *labbook* (see Figs. 2A/B and 3A/D). The *long-term memory storages* allow ACRA to *learn* from previous experiments by providing previously translated examples as well as resolving ambiguities within the context of the prompt (compare SI Section 1 for a detailed description of the workflow). Though ACRA was mostly tested for handling English literature procedures, it showed impressive cross-language capabilities when parsing foreign language documents and procedures, potentially lowering language barriers in scientific reporting. To showcase the potential of ACRA, we demonstrate the automatic parsing, translation, and sanitization capabilities on six reactions and two different *XDL*-integrated platforms. We were able to show that ACRA can autonomously parse synthetic procedures from a diverse set of literature sources, plan, and translate them into *XDL* and finally execute the translated procedures, showing robustness to language ambiguities, different languages, and even potential reporting errors.

## Results
### Extracting synthesis information from chemistry literature

The vast amounts of data contained in scientific publications, regarding compound properties, reactivity, reaction execution, and product analysis can be spread over 10 s to 100 s of pages, typically divided into main publication article and supporting information. While the main publication usually contains a higher-level description of the performed experiments and most relevant results, spectroscopic analysis data, like nuclear magnetic resonance or mass spectrometry, are usually hidden in the supporting information alongside potentially relevant additional information to accurately recreate the experiments.

To harvest this data, the first stage of ACRA (*scraping-agent*) parses a given literature text (and its supporting information or any other documents if provided) and extracts data into a knowledge graph (KG) of relevant synthesis-related information into a predefined structure similar to previous work[2,42,45] (Fig. 3A, D and SI Section 1.2). ACRA parses literature resources by first chunking a given text into 4096 token fragments and then iteratively extracting and combining the data until all text has been parsed (SI Section 1). The *scraping-agent* is instructed to extract all chemical names with their abbreviations and synonyms, procedure texts, purification data, analytical data, and additional information anywhere in the documents (SI Section 1.1.1). Additionally, the initial text document is embedded into a vector database in chunks of 2048 tokens and referenced in the knowledge graph for later retrieval during translation of the procedure, if additional information is required. This way, ACRA can autonomously extract procedure descriptions, chemical information, analytical data, and any pitfalls or limitations highlighted by the authors from vast amounts of text. To test the extraction of synthesis-related information and the construction of the KG, we executed the literature extraction and KG generation module on 20 scientific publications (statistics for 10 shown here) and an organic chemistry PhD thesis (SI Section 2). The cross-language capabilities were tested on an organic chemistry undergraduate practical transcript written in German (SI Section 2.1). In total, 717, 57, and 117 procedures were extracted from the scientific publications, the PhD thesis, and the German practical script, respectively. To estimate how many of those procedures contain all the required information to reproduce a given procedure, ACRAs *procedure-agent* was executed, aiming to resolve any ambiguities and fill in missing data for the unambiguous translation to *XDL* (SI Section 1). During the sanitization, the *procedure-agent* categorizes the extracted procedure into "executable", "blueprint" (general procedure descriptions, etc.), and "incomplete". We manually evaluated the classification on 39 procedures, showing an overall classification accuracy of 67.5% (SI Fig. S23). 75% of failure cases resulted from ambiguously referenced chemicals or references

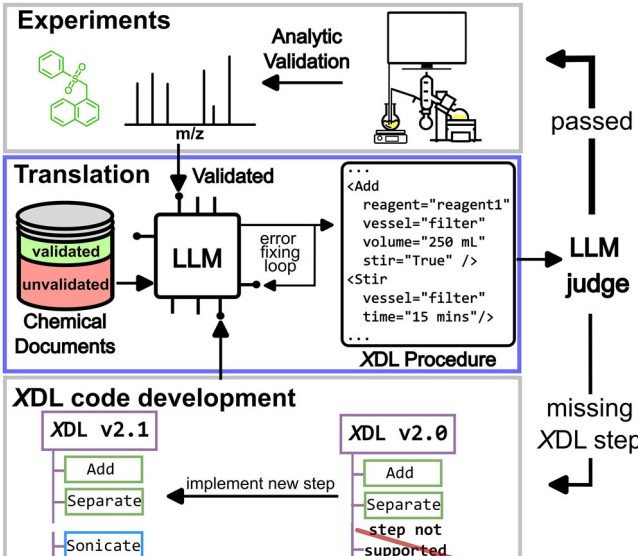

**Fig. 1 | Conceptual overview of the workflow presented in this work for automated synthesis verification and iterative code development on demand of XDL.** Chemical documents are translated by an LLM into XDL procedure code via an error-fixing loop, assessed by an LLM judge, and either passed for analytic experimental validation or returned for iterative XDL code development when unsupported steps are identified.

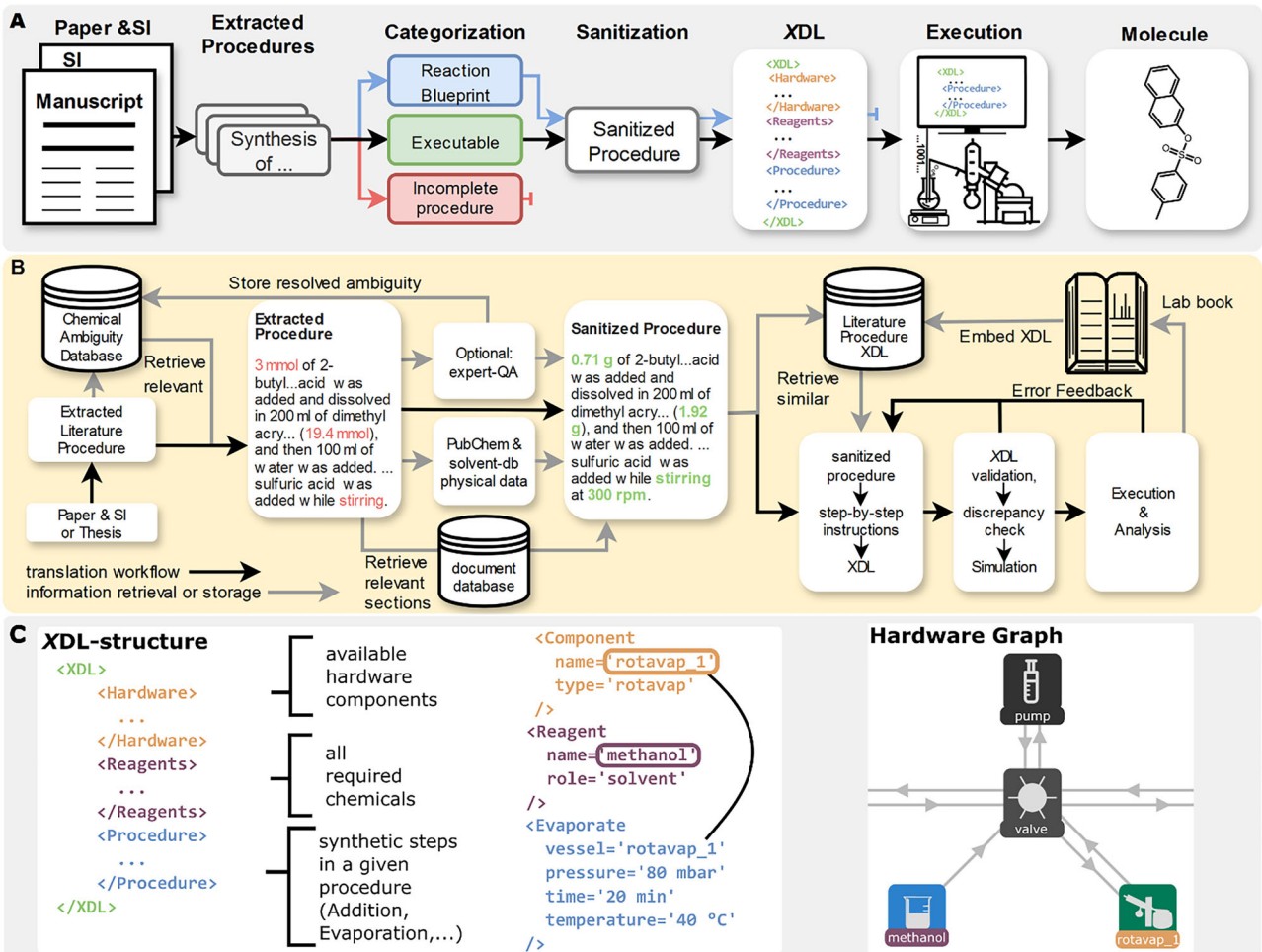

**Fig. 2 | Overview of the proposed framework for automated extraction, translation, and validation of synthesis procedures. A** Simplified overview of the proposed framework for extraction, sanitization, translation and validation of chemical reaction procedures using LLM-based agents. **B** Detailed depiction of the flow of chemical procedures from literature to robotic execution. First, entire papers (and their supporting information/ or any other text document) are parsed by a scraping-agent, and all synthesis-related data is extracted to a knowledge graph. All extracted procedures alongside relevant chemical ambiguities and physicochemical data are passed to a second agent (procedure-agent) to sanitize the procedure (fill in missing physicochemical information, etc.). The sanitized procedure is categorized by the procedure-agent, and subsequently translated by the XDL-agent into XDL. The translated procedure is (if needed) iteratively improved by a three-step sanitization pipeline. Finally, the validated XDL is stored after optional physical execution and analysis, alongside the extracted data, into a labbook. The validated XDL is embedded and stored in a vector database, which is used as a long-term memory of the XDL-agent for future translations. **C** Simplified description of an XDL procedure, and depiction of the hardware graph to execute the given procedure.

to general procedural instructions. Out of the 717 extracted procedures from the scientific publications, 427 were marked as executable, 89 as blueprint, and 201 as containing missing information. The procedures from the German undergraduate practical and PhD thesis were categorized as 48, 2, 7 and 93, 6, 18, in each category, respectively (Fig. 3C). The extraction of synthesis-related data from synthetic procedures and literature sources was assessed based on the identification of chemical entities and the retrieval of analytical data, showing good overall performance (Fig. 3E/F and SI Section 2.2). We additionally compared the extraction of chemical entities against ChemDataExtractor 2.0[45], showing competitive performance (Fig. 3E, SI Section 2.2 for details).

**Precise and executable XDL procedures via validity check, discrepancy analysis, and hardware-constrained simulation**

While the extraction of relevant data from largely unstructured literature procedures and sanitization of procedures are significant steps in automating chemical synthesis validation, accurate translation of literature procedures into unambiguously executable robotic instructions remains a challenge. The Chemical Descriptor Language XDL has been used to

unambiguously and reproducibly execute and share chemical reactions in a hardware-agnostic manner[31,40,41,46,47]. While translation from literature procedures to XDL has been presented before[23,40], a significant bottleneck remains in the accurate translation and automatic validation of these translated procedures. Whereas syntax errors have been used before to iteratively improve upon previously generated XDL instructions[23], we found that a substantial proportion of translated procedures remained syntactically erroneous, had missing steps, and were not sufficiently validated on a realistic robotic setup. To improve upon these shortcomings, we implemented a validation pipeline that first creates a valid XDL with error feedback from an XDL parser that can simultaneously find all syntactic errors in a given XDL. The XDL is then scrutinized by a *critique-agent* instructed to find any discrepancies in steps in the XDL that were mentioned in the literature procedure and implement them subsequently (typically referred to *LLM-as-a-judge*[48]). Finally, the generated XDL is mapped to a predefined robotic platform (Chemputer or Opentrons-OT2), and the execution of these steps is executed in simulation, constrained by the robotic platform (Fig. 4A). The XDL parser captures syntactical issues, ill-defined physical units (e.g., temperature and pressure units), and missing hardware or

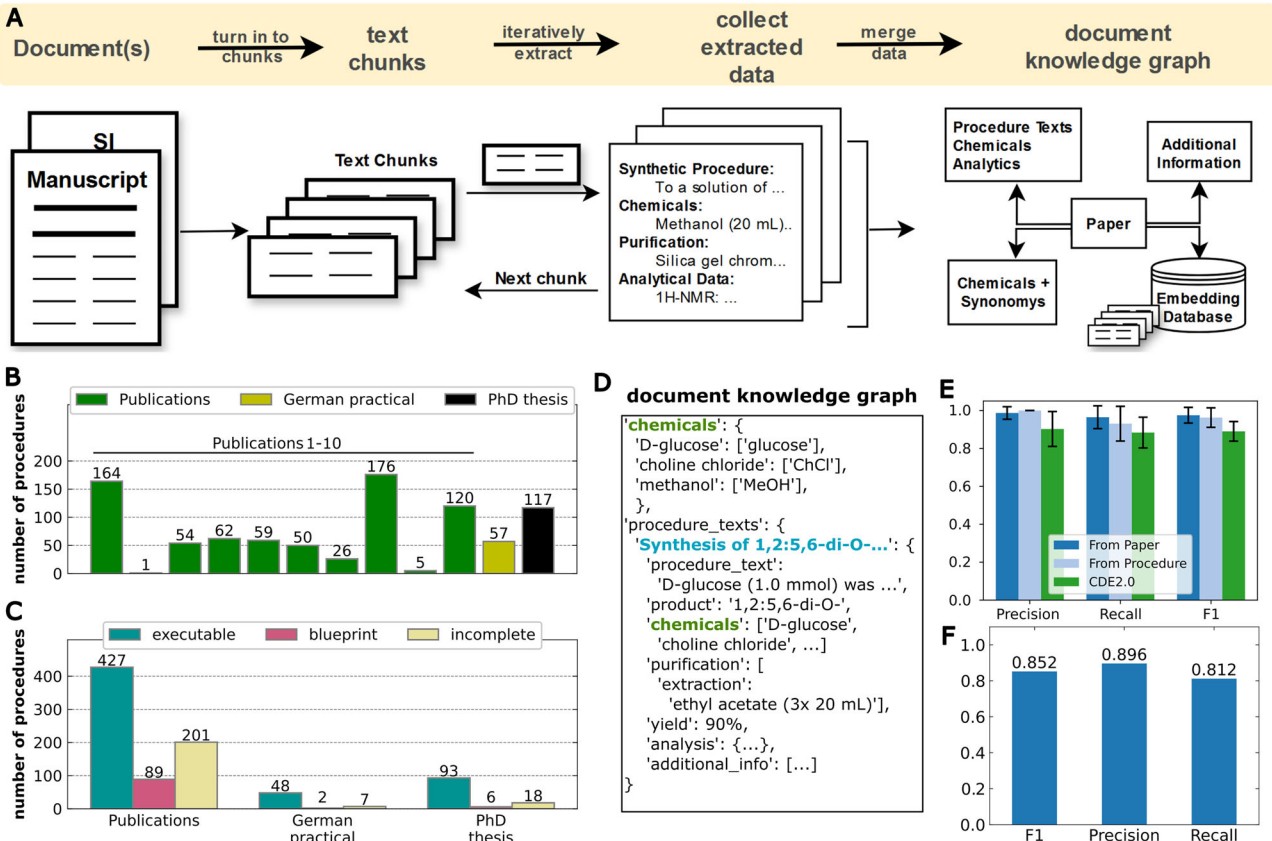

**Fig. 3 | Synthesis data extraction from chemical literature documents. A** Synthesis data is extracted from documents containing chemical synthesis information and turned into a knowledge graph by first extracting all textual data, chunking the text, iteratively extracting synthesis-related data in a predefined JSON format, and finally combining all extracted data into a combined data structure (knowledge graph). The scraping-agent is instructed to extract all chemical names with their abbreviations and synonyms, procedure texts, and purification data, as well as analytical data. Additionally, the initial text document is stored and embedded into a vector database and referenced in the knowledge graph for later retrieval during translation of the procedure, if additional information is required. **B** Benchmark of extracting synthesis procedures from different document types. Ten publications (compare SI 2.1), a script from an undergraduate organic chemistry practical in German language, as well as an organic chemistry PhD thesis, were used to test the extraction capabilities. In total, 717 procedures were extracted from the publications, 57 from the organic chemistry practical transcript, and 117 from the PhD thesis. **C** Categorization of the extracted procedures by the procedure agent into executable, blueprints (general procedures, etc.), and incomplete procedures. **D** Simplified depiction of a document knowledge graph. **E** Precision, Recall, and F1 scores for the extraction of chemical entities from eight synthetic procedures or papers (SI Section 2.2), and comparison, to ChemDataExtractor 2.0 (CDE2.0). **F** Precision, Recall, and F1 scores for the extraction of analytical data from 20 procedures (SI Section 2.2). Error bars represent the standard deviation.

chemical reagents. Simulation of the physical execution was tested via computational simulation of the produced procedure, capturing errors such as invalid temperature or rotation speed ranges. While these verification measures help confirm the executability of the translated procedure, they do not guarantee its completeness or accuracy. Importantly, the *critique-agent* (SI Section 1.1.5) proves to be a vital part of the accurate translation of syntactic procedures, identifying missing or disordered steps. Examples for each of the three-stage feedback responses are shown in Fig. 4B (SI Section 1.3 for details). To translate literature procedures into *XDL*, the *XDL*-agent (SI Section 1.1) was instructed to first extract all chemicals and their role in the procedure (e.g., solvent or catalyst), then decompose the procedure into step-by-step instructions and translate them into *XDL* in a *ReAct-style* response format[49], and finally combine the individual steps into a single *XDL*, within a single prompt. During the iterative improvement, the *XDL*-agent is instructed to first map the errors identified from the validation stages to the part of the *XDL* procedure causing the error, and finally correct the corresponding lines. In each iteration of the translation, the five most similar synthetic procedure-*XDL* pairs are provided within the prompt, allowing to *XDL*-agent to learn from previously translated and validated procedures (see below and SI Section 1.2). Additionally, the *XDL* documentation and previously resolved ambiguities are provided within the context of the prompt (SI Sections 1.1, 1.2, and below).

The translation into *XDL* was manually validated using extracted synthetic procedures and direct translation from literature sources, demonstrating strong performance across all evaluated metrics compared to previously established methods (Fig. 4D, ACRA$_{paper}$ and ACRA$_{procedure}$; SI Section 2.3 for details). Additionally, the effectiveness of the *critique-agent* was demonstrated by ablation (ACRA$_{procedure-no-judge}$ in Fig. 4D). Finally, the performance of the validation stages was examined with 150 procedures from the ChemRnD dataset[23] (three times 50 independently sampled; SI Section 2.4). The maximum number of iterations (*XDL*-generation → Validation → Feedback → *XDL*-generation) for the generation of error-free *XDL* after all three validation stages was set to 6 (see SI Fig. S21 for distribution of actual number of iterations). 99.33% of procedures were translated into valid *XDL* (passing the *XDL*-validity check), and 94.67% of procedures were additionally successfully validated during procedure-*XDL* discrepancy check and simulation of the execution, ensuring their accuracy and executability on a suitable platform (Fig. 4C). This highlights the necessity for validation beyond the mostly syntactical validation presented in previous studies to generate accurate and executable *XDL* procedures. We additionally tested two frontier open-source language models with up to 70b parameters as the model behind the *XDL* agent. Neither of the models generated valid *XDL*, however (compare SI Section 2.6 for an output comparison).

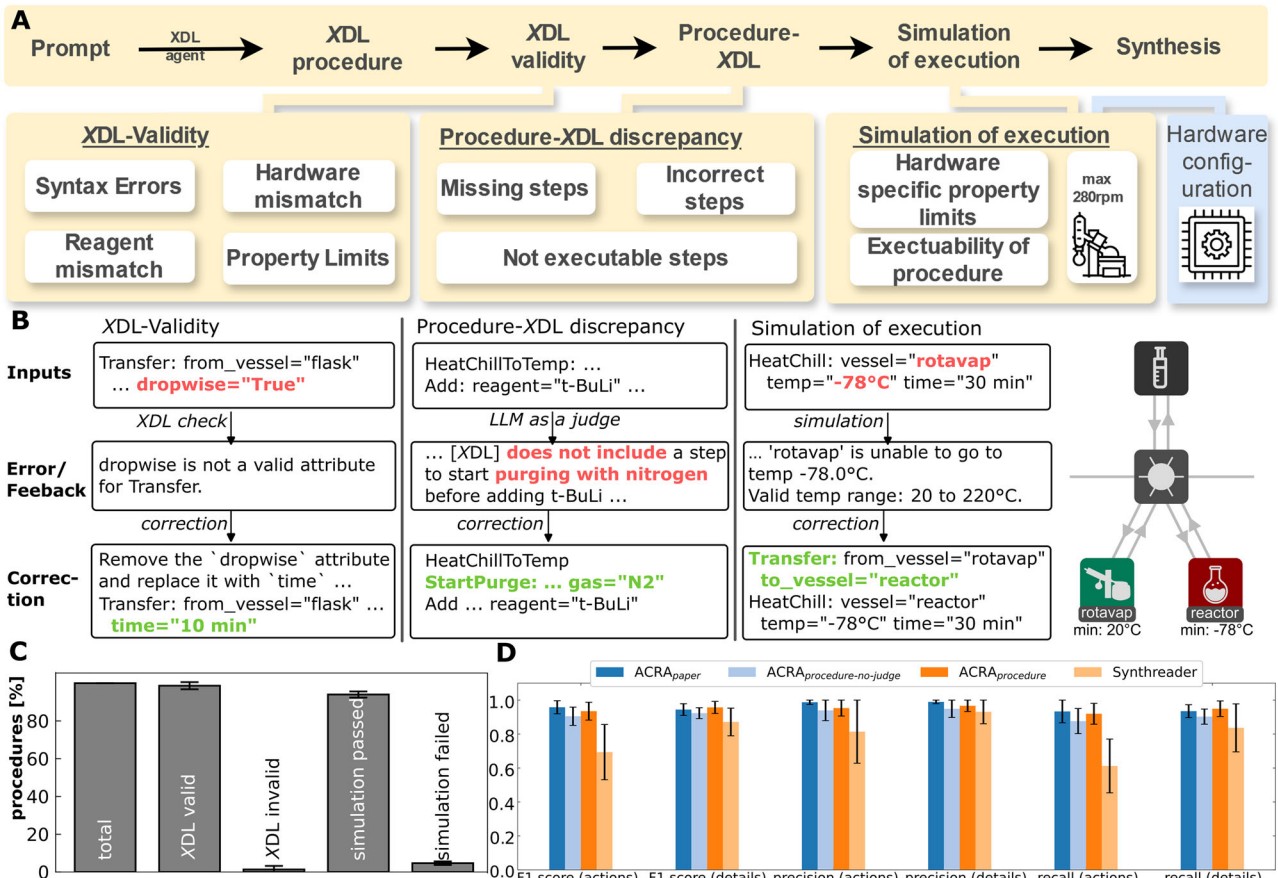

**Fig. 4 | Generating accurate and executable XDL procedures for automatic synthesis execution. A** Overview of workflow from synthesis procedure to validated XDL and synthesis. The procedure is translated into XDL, checked for errors, analysed to find discrepancies between the natural language procedure and the XDL, and finally simulated in a hardware-constrained environment. All errors captured along this pipeline are iteratively fed back to the LLM-agent to correct the errors. **B** Examples of the three stages in which the XDL is scrutinized. **C** Statistics of a total of 150 randomly selected procedures (three times 50 independently sampled), on passing the stages described in (**A**). **D** Benchmark results for the translation of 8 synthetic procedures into XDL (compare SI Section 2.3 for details) compared to previously published work[40]. Error bars represented the standard deviation.

## Using long-term memory storage and past experiments to learn from experience

Though reported synthetic procedures usually contain sufficient information for an expert chemist to infer implicitly assumed information, autonomously identifying missing or assumed information can pose a significant problem in autonomous synthesis execution. To tackle this problem, we equipped ACRA with an ambiguity database containing implicit knowledge of expert chemists identified in previously published literature procedures. To initialize the database, 5 synthetic procedures were carefully annotated and described, and each piece of the procedure, with its detailed explanation, was stored in a vector database (Chemical Ambiguity Database; compare SI Section 1.2.1).

The explained segments of the procedures were encoded into a 2048-dimensional vector using OpenAI's *text-embedding-large* model, to enable semantic search for further clarification. During the sanitization of procedures, relevant parts from the Chemical Ambiguity Database (CAD) serve as a long-term memory containing chemical intuition or implicit information that is usually not explicitly captured in textual form and learned by synthetic chemists during their educational programs. Relevant information contained in the CAD is retrieved by embedding sentences of a procedure into the same 2048-dimensional vector space, and the semantically most similar information is selected as estimated by the cosine similarity of their embedding vectors. This information is then included within the context of a prompt (SI Section 1.1). Additionally, ACRA can optionally ask questions about any parts of a given procedure to an expert chemist, whose answer, alongside the section of the procedure in question, is stored in the long-term

ambiguity database available for subsequent executions (SI Sections 1.1 and 1.2). This information together with molecular information extracted from PubChem about the chemicals used in each procedure (i.e., conversion of chemical names to IUPAC, and g/mmol to g/mol) and a local solvent database (containing boiling points etc.), is then provided to the *procedure-agent* to prepare a procedure with all necessary information to be translated into XDL (SI Section 1 for details). During the translation of sanitized procedures into XDL, the 5 most similar previously translated synthetic procedures and the corresponding XDLs are provided within the context of the prompt to help translation, usually referred to as *few-shot prompting* via *Retrieval Augmented Generation* (RAG; SI Section 1.2).

To evaluate the influence of the *memory* and data components, 75 procedures were translated into XDL with different parts of the workflow ablated. First, to fully test ACRAs capability to translate synthetic procedures to XDL, we initialized the XDL database with 62 procedures of previously published synthetic procedures-XDL pairs[47]. Each successfully translated procedure-XDL pair was added to this database and is thus available for subsequent translations. We found that ACRA successfully translated 100% of the procedures into valid XDL (XDL validity check), and 6.7% failed to pass at the later stages (discrepancy check and simulation of execution). To test the influence of removing the XDL database, the procedures were additionally translated into XDL once, without any initial XDLs in the database but with continuous addition of successful translations, and once completely omitting the XDL database. While in both cases, 100% of the procedures could be translated into valid XDL, 9.3% and 10.7% of procedures did not pass the latter two stages of the validation pipeline,

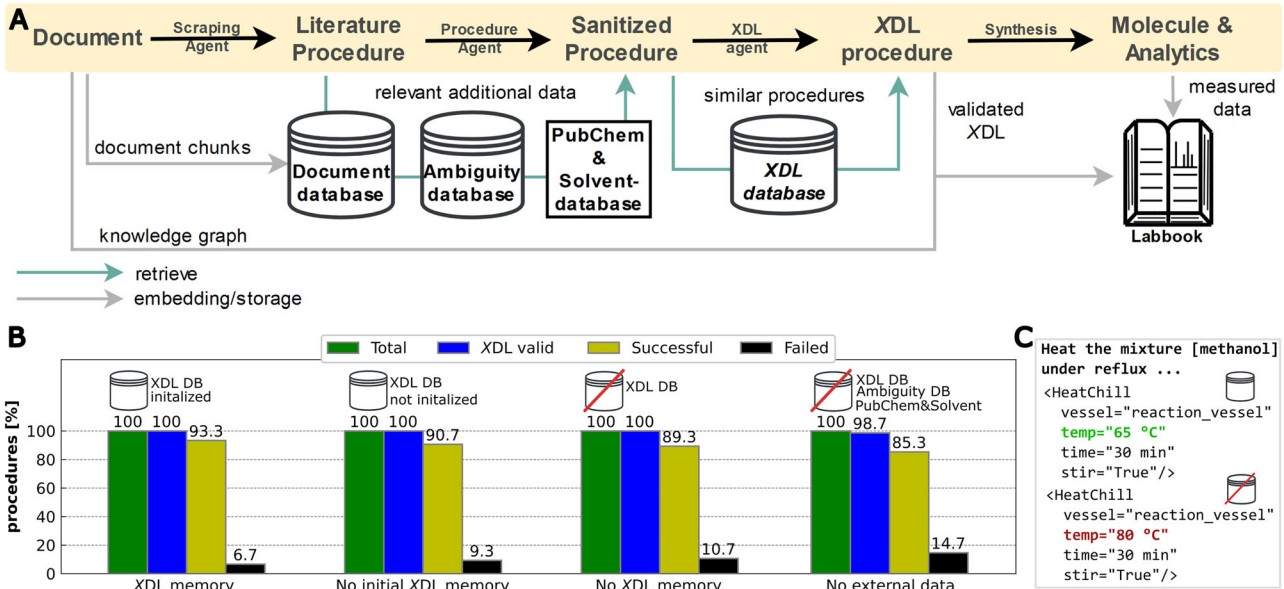

**Fig. 5 | Overview of inclusion of external data sources and long-term memory components in the translation of synthetic procedures to XDL. A** Given a document containing the description of a synthetic procedure, the document is embedded in chunks of 2048 tokens into a vector database. During translation of extracted procedures, ACRA is instructed to ask questions that may be present in the document, but not the procedure (e.g., general procedure instructions), which are then retrieved from the vector database and included in the prompt. Additionally, previously resolved ambiguities are included, as well as physicochemical information about the identified chemicals in the procedure. During the XDL generation, the 5 most similar, previously validated XDLs are retrieved from a vector database and included within the prompt. The combined data from the original document, a reference to the document database, resolved ambiguities, the validated XDL, and any analytical data from the original procedure are finally stored within a virtual labbook. **B** Influence of inclusion (or exclusion) of different storage and external data on the translation success of literature procedures to XDL. While in all stages, valid XDL was generated in almost 100% of cases, the overall rate of success increased from 85.3% to 93.3% with the inclusion of the different data and memory sources. A total of 75 randomly sampled procedures from the ChemRnD dataset were translated per run. **C** Translation example where exclusion of data sources leads to a validated XDL, but high temperature for refluxing of a methanol-based solution.

respectively (Fig. 5B). In a last test, all databases and external data sources were removed from the translation process.

This way, no XDL database, no Chemical Ambiguity Database, and no additional chemical information was provided for the translation. The document database will only be used if the translation starts from a document, and was thus not used by default in this experiment. While 98.7% of all procedures could still be translated into valid XDL, 14.7% of procedures failed in one of the latter stages (discrepancy check or simulation of execution). This shows how the use of previously translated and validated examples can be used to systematically improve the translation capabilities of LLM-based *agents* in chemistry without the need for any further training of the underlying models. The use of additional external sources, like a curated chemical database, proves to be a valuable addition to increasing the success rate of translated procedures.

### Systematically improving XDL by identifying non-executable synthetic steps

Though a wide variety of reactions have been demonstrated to be executable using XDL procedures, XDL is still actively developed to add support for an increasing number of synthetic operations. Additionally, while most newly published procedures will not contain logically new steps, synthetic chemistry itself is changing and thus might require new synthetic steps. To identify and suggest new steps that should be added to the XDL standard, the *critique-agent* described above was instructed to identify missing steps and steps currently not executable within XDL. Steps that were identified as not executable were collected, clustered, and analysed to provide suggestions for new XDL steps (Fig. 6A, B). Suggested XDL steps can then be analysed and finally integrated based on urgency and ease of integration for the next version of XDL. To establish a roadmap for improving XDL and enhancing its universal applicability, 350 steps identified as non-executable were analysed based on data collected throughout this study. The steps were clustered, resulting in 26 new feature suggestions for future generations of XDL

(Fig. 6C and SI Section 4), showcasing how this workflow can be used to systematically identify relevant new step suggestions. Additionally, 65 million procedures from the Reaxys database were analysed for synthetic keywords identified and provided within the database.

The 200 most frequently identified keywords were analysed and grouped into 20 categories (see SI Section 4.3). Categories for which no abstract XDL step exists in the current version of XDL are shown in Fig. 6D. Comparison of the newly suggested steps as an outcome of the not executable steps from the *critique-agent* and those from keywords from Reaxys procedures shows that the not executable steps result in substantially more specific suggestions, ranging from new attributes (e.g., rate-control for heating or cooling steps) to steps requiring new hardware implementations (e.g., automated TLC analysis). The 26 new XDL feature suggestions are grouped by type of adjustment required for implementation and ranked by ease of implementation and urgency, resulting in a suggested roadmap for future XDL generations (Fig. 6D).

At the software level, implementing new XDL steps requires propagation across a multi-layered architecture, from the hardware-independent XDL parser defining cross-platform syntax, through hardware-specific libraries encoding platform-specific execution logic. Implementation of new hardware is mediated through a corresponding implementation stack of low-level libraries providing standardized communication protocols for commercial or custom devices. This architecture facilitates modular expansion but requires each new module to conform to these layered application programming interface standards for cross-platform interoperability[40,50].

### Paper to molecule

To showcase the capabilities of ACRA in helping to automate chemical synthetic literature validation from parsing a literature source to execution of a synthetic procedure, we tested it on two English synthetic procedures, one German synthetic procedure, as well as one scientific publication, covering a variety of document types and use cases on a Chemputer

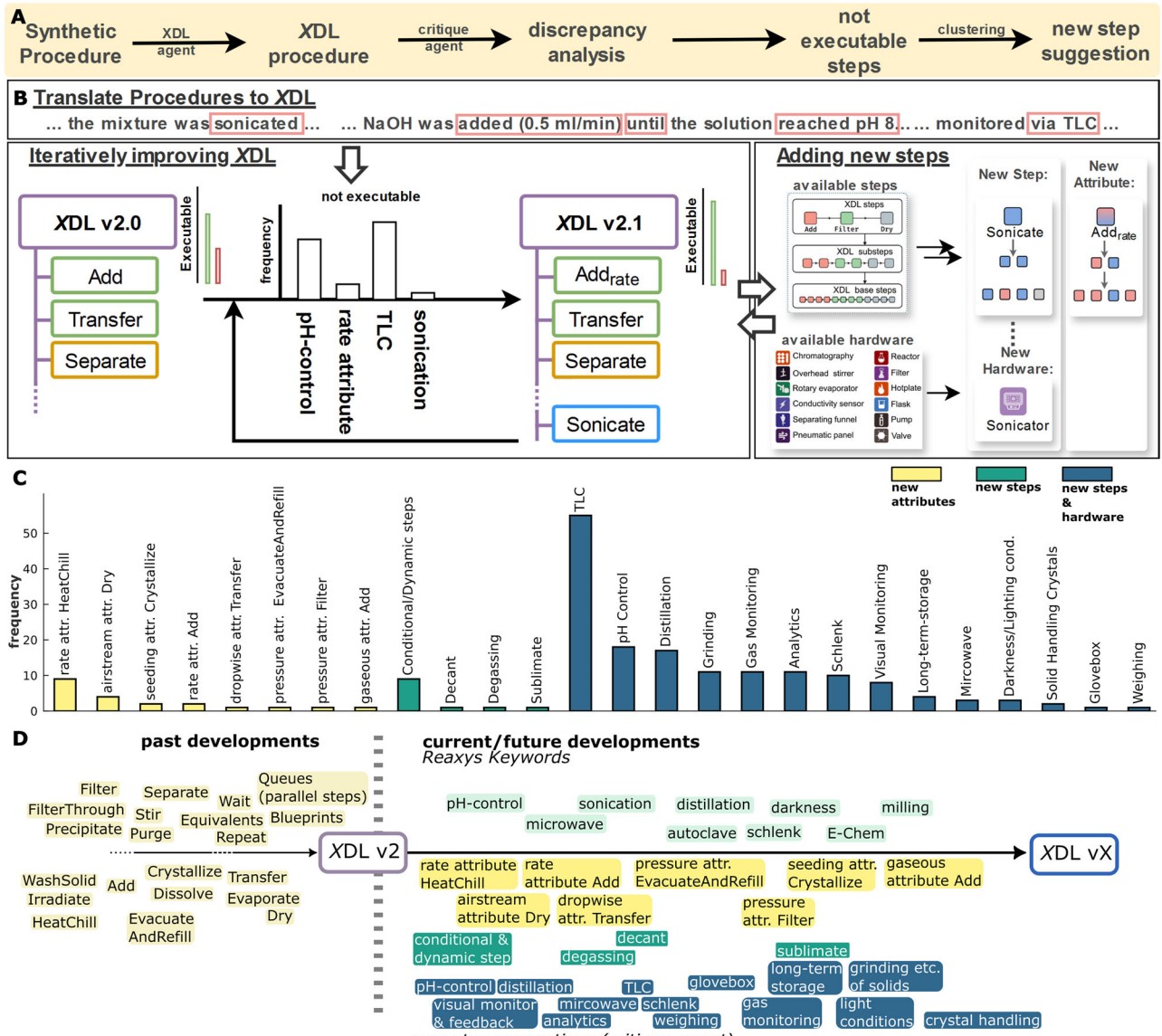

**Fig. 6 | Capturing currently unsupported steps from synthetic procedures and creating a roadmap to systematically improve XDL. A** Developed workflow to capture steps that are not executable in the current version of XDL. Synthetic procedures get translated into XDL as described above. The critique-agent then checks the translated procedure and the original procedure and captures steps that can currently not be translated into XDL steps. The combined, non-executable steps from a set of translated procedures are then clustered and used to plan new steps for future versions of XDL. **B** Conceptual overview of how non-executable steps are used to iteratively improve XDL. Not executable steps from a batch of procedures are first clustered, categorized, and subsequently, depending on urgency and ease of implementation, included in the XDL language. Depending on the update, this results in a new attribute or requires a new step and hardware integration. Part adapted from literature[40]. **C** Classification of not executable steps in this work into potential new XDL steps or features grouped by type of modification required for implementation (new attribute, steps, or step and hardware support). **D** Overview of XDL development and supported steps. The 26 newly suggested steps are grouped by type and sorted by urgency and ease of implementation from left to right. Additionally, to the non-executable steps identified as specified above, ~65 million procedures from the Reaxys database were analysed on their most frequent keywords, and the top 1000 keywords were clustered to yield new step suggestions.

platform (Fig. 7A–E and SI Section 5.1–5.4). We furthermore show how the use of standardized synthetic procedure generation via XDL allows for simple adaptation of our workflow to cross-platform execution[41] with a limited set of supported XDL steps and two additional synthetic procedures (Fig. 7F/G and SI Sections 5.5/6). The two English procedures detailed the synthesis of p-toluenesulfonate[51] and 2-Methyl-2-(3-oxopentyl)-1,3-cyclohexanedione[52] (Fig. 7B/C) and were translated into 23 and 20-step XDL procedures, respectively. In the first, refluxing temperatures were correctly assigned, and volumes for adjustment of pH were estimated since no adjustment of the pH adjustment step was implemented in XDL version for this study. ACRA notably labeled the adjustment of the pH as a not precisely executable step. The German synthetic procedure described the synthesis of 3-Methoxy-3-oxopropanoic acid. During the translation, all chemical names, as well as the synthetic procedure, were correctly translated. Additionally, the procedure specified lifting the reaction vessel halfway from the oil bath, which is not a directly executable step in XDL. ACRA determined that the reaction vessel could not be "lifted from the heating vessel" and instead adjusted the temperature from 65 °C to 50 °C, effectively aligning with the intended purpose of the step. To further showcase the potential of automatic execution of literature procedures, ACRA was provided with a scientific publication detailing the synthesis of multiple sugar compounds. Five synthetic procedures were extracted from the procedure. Two of the procedures were classified as blueprints (they were general procedures), while the other three were correctly classified as executable.

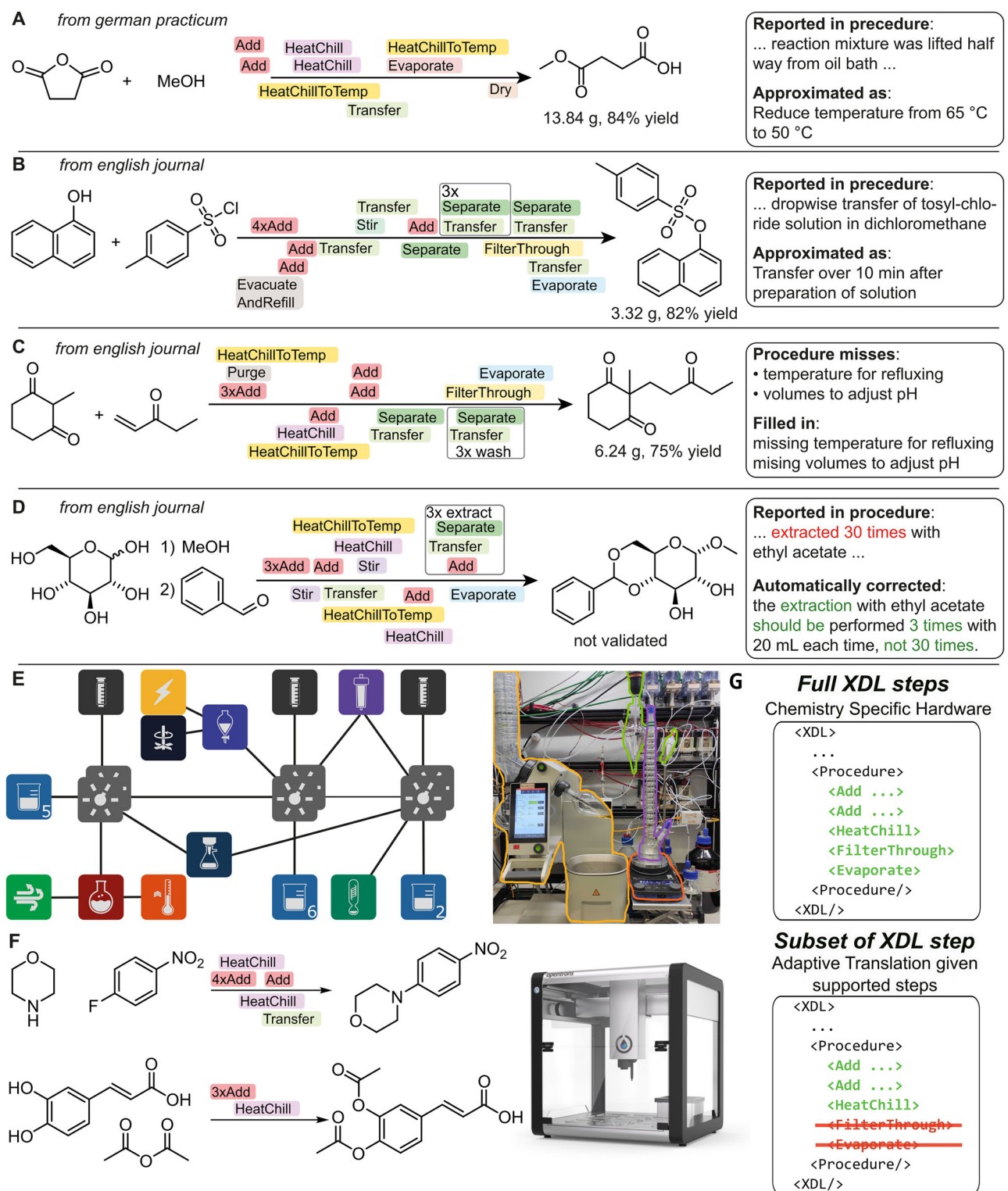

One of the procedures was selected for validation, describing the synthesis of Methyl 4,6-O-benzylidene-α-D-glucopyranoside[53]. Notably, the procedure mentioned extracting 30 times with 20 ml ethyl acetate, which appears to be reporting an error in the procedure. During the translation, ACRA identified this to be changed to 3 instead, given the context of the paper and previously translated examples, without further instructions. The translated procedure was executed, but no conversion of starting materials could be detected. To verify that the missing conversion was not the result of a false correction, the synthesis was additionally executed with varying hardware

implementations by a synthetic chemist closely following the reported procedure. Nevertheless, the reported reaction could not be verified without substantial alterations to the procedure and was thus classified as not reproducible, highlighting the potential of the presented system. The synthesis of 4-(4-nitrophenyl)morpholine and (2E)-3-[3,4-bis(acetyloxy) phenyl]-2-propenoic acid (Fig. 7F) was performed on a commercial Opentrons-OT2 platform. To showcase the adaptability of our approach, the provided *XDL* documentation was restricted to a limited set of instructions previously implemented for this platform[41] (Fig. 7G). Both

**Fig. 7 | Synthesized molecules from XDL procedures generated by ACRA on a chemputer and opentrons-OT2 platform. A** Synthesis of 3-Methoxy-3-oxopropanoic acid. A procedure in German for the synthesis of 3-Methoxy-3-oxopropanoic acid was provided to ACRA, automatically translated and adapted to be executable on a Chemputer platform (SI Section 5.1). **B** Synthesis of p-toluenesulfonate via a 23-step XDL procedure generated from a literature procedure (SI Section 5.2). Notably, the dropwise transfer of solutions was approximated as a transfer of 10 min. **C** Synthesis of 2-Methyl-2-(3-oxopentyl)-1,3-cyclohexanedione from a literature procedure (SI Section 5.3). Temperatures for refluxing were correctly assigned, and volumes to adjust the pH were estimated, pH-control was not available on the given physical platform. **D** Synthesis procedure of methyl 4,6-O-benzylidene-α-D-glucopyranoside directly parsed from a research article detailing the synthesis procedure via a 22-step XDL procedure. Notably, the

procedure mentioned extracting the reaction mixture unusually many times (30 times), which ACRA automatically changed to three times, given the context of the paper (SI Section 5.4). The procedure was executed three times with various cautions hardware implementations and was concluded to be not reproducible without significant alteration of the procedure/including more details in original procedure. For more details confer SI Section 5.4. **E** Example of a Chemputer graph connectivity for one of the experimentally validated syntheses and photo of Chemputer used for experimental XDL validation. **F** Synthesis of 4-(4-nitrophenyl)morpholine and (2E)-3-[3,4-bis(acetyloxy)phenyl]-2-propenoic acid on the Opentrons-2 platform. All yields are based on crude isolated products, whose purity was checked by NMR (see SI Section 5.). Image of the Opentrons OT2 is reprinted with permission[59]. **G** Conceptual depiction of the limited XDL instruction set provided for the translation of synthetic procedures into XDL to be executed on the Opentrons-2 platform.

reactions showed good yield, validating the simplicity of adapting ACRA to new instruction sets.

As evident from the validated examples, the qualitative nature of reported language necessitates a "best guess" approximation to define executable parameters (e.g., "dropwise transfer" to a 10-min duration). Since this is an inherent problem of how chemical procedures are reported, we conclude that the provided experimental validation suggests that these system-derived approximations are chemically reasonable and effectively resolve the inherent ambiguities of the source text in most cases.

## Conclusion

In this work, we demonstrated how LLMs can be used to autonomously validate chemical synthesis literature from parsing of a literature text to final execution of the synthesis. While this approach could be extended to parse non-textual data[54–56], we focused our approach to only parse textual components, leaving the integration of non-textual data for future work. Importantly, we showed how syntactic validation of generated protocols for the robotic execution of synthesis procedures in the XDL is not sufficient for the accurate translation of literature procedures. We demonstrated that a hardware-constrained simulation can be used to generate XDL procedures that are reliably executable. While this work serves as a thorough proof of concept showcasing the potential for integrating LLMs into autonomous chemical synthesis, we hypothesize that such frameworks can be expanded to incorporate more simulation data, eventually enabling LLM-based agents to orchestrate entire laboratories.

Similarly to how a human chemist might learn from previous examples, we showcase how integrating previous experiences into the translation workflow helps to substantially improve the translation success rate. Although the CAD was valuable (compare Fig. 5), its need for extensive manual curation limits practical expansion, even though the current data already resolves the most common pitfalls we initially observed. Nonetheless, expanding on this concept might thus be an important step into open-ended chemical discovery. This requires an unambiguous and universal way of representing chemical procedures (compare Fig. 7E). To improve on existing paradigms, we showcase how feedback from an LLM can be used to systematically suggest missing operations (here XDL steps) to fully cover all operations required to perform chemical experiments. We identified 26 new XDL feature suggestions and created a potential roadmap for future implementation. However, it is important to highlight that challenges in fully autonomous execution of scientific literature still face experimental challenges, such as automated purification of unknown mixtures, automated analytics, or even reagent availability, though some solutions have been proposed[26,57,58]. Robust solutions to these challenges allow to extend LLM-controlled autonomous systems to execute many-step, successive synthesis procedures.

## Methods

The *critique-*, *XDL-*, and *procedure-agents* used GPT4o for all prompts. Only the *scraping-agent* used GPT4o-mini for the reduced cost and increased token-output size. All *embeddings* were generated with the OpenAI model *text-embedding-large*. All code was written in Python 3 with standard

libraries, apart from the XDL and Chemputer-specific libraries. The default XDL library (https://gitlab.com/croningroup/chemputer/xdl) was modified to capture all syntactic errors in parallel. The ChemputerXDL was modified to automatically map generated XDLs to a predefined hardware graph. XDL files (.xdl) and Chemputer graph files (.json) can be viewed and edited with the ChemIDE app on https://croningroup.gitlab.io/chemputer/xdlapp/. The XDL software standard is linked here: https://croningroup.gitlab.io/chemputer/xdl/standard/index.html. All XDL and Chemputer-specific software packages can be made available upon reasonable request.

## Data availability

All benchmark data are available at https://doi.org/10.5281/zenodo.18980981. Any additional information required is available from the lead contact upon reasonable request.

## Code availability

The code written for the implementation of ACRA is available at https://doi.org/10.5281/zenodo.18980238 or https://github.com/croningp/acra.

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

## Acknowledgements

We acknowledge financial support from the John Templeton Foundation (grant nos. 61184 and 62231), the Gates Foundation (project no. INV-058957), the Engineering and Physical Sciences Research Council (EPSRC) (grant nos. EP/L023652/1, EP/R01308X/1, EP/S019472/1, and EP/P00153X/1), the Breakthrough Prize Foundation and NASA (Agnostic Biosignatures award no. 80NSSC18K1140), MINECO (project CTQ2017-87392-P), EC 101046836 CATART, and the European Research Council (ERC) (project 670467 SMART-POM). We like to acknowledge Dr Dean Thomas for the feedback on the manuscript and many helpful discussions.

## Author contributions

L.C. conceived the idea and research plan together with S.P. and M.J. S.P. built and developed the workflow for the implementation of ACRA module, with contribution of M.J. S.P. and M.J. implemented the experimental setup. M.J. and L.C. mentored S.P. S.P. wrote the manuscript with contributions from all authors.

## Competing interests

The authors declare no competing interests.
