## [Transparent Peer Review file · Communications Chemistry]

Verification and Execution of the Scientific Literature via Chemputation Augmented by Large Language Models

Corresponding Author: Professor Leroy Cronin

Version 0:

Reviewer comments:

Reviewer #1

(Remarks to the Author)

This manuscript reports a large language model (LLM)-driven multi-agent system (ACRA) that automates the extraction, translation, and robotic execution of chemical synthesis procedures from literature. The study integrates language models with an automated chemistry platform (Chemputation), representing an important step toward realizing the concept of a "self-driving laboratory." The authors extracted information from 21 papers of different languages and sources, and successfully reproduced six reactions on automated experimental platforms. Multiple agents were introduced throughout the workflow, responsible for validation, simulation, and knowledge accumulation. Overall, the framework is conceptually clear and structurally complete. However, the manuscript still presents several critical issues regarding novelty, comparative evaluation, extraction robustness, literature diversity, and experimental generalizability, which require clarification and additional data before this work can be considered for publication.

Q1: While the paper emphasizes "end-to-end automation" as its main contribution, existing studies (e.g., Boiko et al., Nature 2023; Bran et al., Nat Mach Intell 2024) have also explored using LLMs for chemical experiment automation. The current manuscript does not clearly demonstrate ACRA's advantages or performance gains relative to existing systems. The authors should provide a comparative analysis.

Q2: The description of the automated experimental platforms is overly brief, and the Supplementary Information also lacks technical details. The authors should elaborate on how automated syntheses were executed in practice to improve reproducibility and transparency.

Q3: Although several successful demonstrations are presented, many literature procedures could not be reproduced, especially those requiring TLC steps, which frequently appear in the failed cases. The authors should provide a systematic analysis of these unexecuted steps and failure modes, identifying underlying causes and suggesting possible solutions.

Q4: The Critique-Agent employs an LLM-as-a-judge approach. Are there instances of misjudgment or oversight in its decisions? Are there user confirmation or rollback mechanisms in place?

Q5: The overall performance of the "literature-to-experiment" pipeline could be more clearly quantified. The authors are encouraged to report success rates at each stage to demonstrate the specific contribution of each agent and the collaborative workflow.

Reviewer #2

(Remarks to the Author)

The authors present an ambitious study that introduces Autonomous Chemputer Reaction Agents (ACRA), an LLM-based multi-agent workflow designed to extract chemical procedures from literature and translate them into the XDL chemical programming language.

The work is timely and introduces several interesting aspects such as:

- A three-stage validation pipeline (syntactic check, procedure-XDL discrepancy analysis, and hardware-constrained simulation) for extracting and correcting chemical procedures.
- A memory option to store current knowledge of chemical procedures combined with an "LLM-as-a-judge" approach to

detect and correct discrepancies in chemical procedures.

- A method to expand XDL scope by extracting not-yet-implemented procedure steps.
- Some actual syntheses to test the approach in real life.

I would like to compliment the authors on the well-conceived and thoughtfully executed project. The manuscript is promising, and I think it will be suitable for publication after some minor revisions aimed at contextualizing the authors findings.

1. While the authors have clearly put a lot of thought and effort into this manuscript, I believe the work should be framed as a (very thorough) proof of concept. Stating this more clearly in the manuscript would help set the right expectations for the reader. I suggest adding a sentence in these regards in either the abstract, introduction, or conclusion.
2. The authors present their findings in a positive light, and I agree that their results are very encouraging. At the same time, it would be interesting to know more about the challenges and what isn't working so well yet. For example, the authors could discuss the system's current readiness for multi-day or multi-step syntheses and give readers a better idea of the future improvements they envision (beyond just extending to non-textual data).
3. The authors should discuss the classification accuracy of 67.5% in more detail, as this seems to me to be a major bottleneck. In my view, the manuscript doesn't fully address the implications of this performance level. I also think the authors should better describe how a misclassification (e.g., an 'executable' procedure being flagged as 'incomplete') is handled downstream in the pipeline.
4. Along the lines of performance metrics, I would be curious to know if the authors considered comparing their extraction performance with other published tools (ChemDataExtractor, which is able to handle multimodal input to some extent). They should provide a reason why a comparison to one of these tools wasn't included here.
5. The "Chemical Ambiguity Database" (CAD) is a great addition for providing "expert chemical intuition." The study mentions it is initialized based on just 5 annotated procedures, which is separate from the XDL Database used for few-shot examples. To me, this raises a question about the scalability of this process. I think the authors could add a brief discussion about this, considering whether it might be desirable to have a very large ambiguity database in the future, or if the current approach is a good enough trade-off to keep the workflow lightweight.
6. I think the authors should elaborate more on the "autocorrection" feature, such as changing 30 extractions to 3 in Figure 7D. This is a powerful but potentially risky capability. It would be interesting to discuss how the system distinguishes a reporting error from a genuine (though unusual) procedure and what safeguards are envisioned to prevent incorrect "fixes".
7. Along the same lines, the workflow makes quantitative approximations for ambiguous phrases (e.g., "dropwise transfer" becomes "10 min" or "lifted half way" becomes a 15°C temperature drop). The authors should clarify how these specific values are reasoned.
8. The authors could reframe the failed synthesis in Figure 7D. In my view, the failure to reproduce this reaction is a success for a verification-focused workflow. It's a demonstration of ACRA's utility in identifying non-reproducible procedures from the literature, and a (cautious) statement in this direction could be added in my opinion.
9. The authors mention in the SI (Section 2.6) that open-source models failed to generate valid XDLs. This is an important finding for the readership. I suggest the authors briefly move this result into the main manuscript's discussion, as it highlights the workflow's current dependence on proprietary models.

Reviewer #3

(Remarks to the Author)

Summary

This article by Cronin's group introduces a novel approach to extracting organic reactions into XDL language. An interesting innovation is the use of a database of verified reactions to improve the process. While the article proposes interesting directions, it is challenging to fully assess its scientific rigor. The concept is innovative, but the presentation is convoluted with unclear parameter choices, insufficient data justification, and poorly explained figures.

Major Points

Unclear Optimization and Benchmarking

The article does not adequately explain how various parameters (e.g., chunk size, data splitting) were optimized. For instance, it reports benchmark data but fails to specify if these datasets are representative, optimized, or simply test data from previous work. This raises concerns that the results may not be reproducible or rigorously tested. Figure captions showing benchmark results should specify the support set size and how error bars were obtained.

Biased and Incomplete Datasets

The evaluation datasets appear highly biased, with only eight procedures used for the knowledge graph extraction part. No

detailed scoring or criteria are provided for how tolerances were set (e.g., when temperatures are deemed equivalent), leading to potential inaccuracies in the extracted data. This may result in skewed performance metrics that don't reflect real-world applicability.

Issues with Supplied Data/Code

The supplied Dropbox is extensive but poorly organized, lacking a proper README file or clear instructions. This makes it challenging to navigate the data, as some folders are empty, and log files contain inconsistencies (e.g., a product mentioned that isn't in the extraction). The absence of detailed scores for the knowledge graph part further complicates assessment of the method's effectiveness.

Version 1:

Reviewer comments:

Reviewer #1

(Remarks to the Author)

The authors addressed the previously raised comments. The authors also addressed the reviewer #2's comments.

One more minor comments:

The authors provided a systematic statistical analysis of the non-executable steps in Figure 6 and propose feature suggestions that could motivate the introduction of new XDL steps, which is of clear engineering value. However, it would be helpful if the authors could further elaborate on the flexibility and practicality of implementing these new steps, both at the software and hardware levels.

Reviewer #3

(Remarks to the Author)

The authors addressed all my concerns in a satisfactory manner.

Reviewer questions are shown in *italics*; responses are in normal font.

Response to Reviewer 1

Q1. While the paper emphasizes "end-to-end automation" as its main contribution, existing studies (e.g., Boiko et al., Nature 2023; Bran et al., Nat Mach Intell 2024) have also explored using LLMs for chemical experiment automation. The current manuscript does not clearly demonstrate ACRA's advantages or performance gains relative to existing systems. The authors should provide a comparative analysis.

We thank the reviewer for this critical observation and discussed both studies in our introduction of course. We agree that recent works, particularly Boiko et al. (*Nature*, 2023) and Bran et al. (*Nat. Mach. Intell.*, 2024), have made significant strides in LLM-driven chemistry. However, ACRA addresses a different set of challenges focused on the faithful reproduction of unstructured literature and hardware-agnostic standardization, rather than experiment planning or optimization from scratch.

Boiko et al. Coscientist excels at planning synthesis routes via web search and interacting with diverse APIs (e.g., Opentrons, Emerald Cloud Lab), and Bran *et. al.* Chemcrow in general-purpose tool augmentation use, ACRA is designed to bridge the gap between static scientific documents and universal execution.

We extended our manuscript to better represent these differences and position our work within the literature:

pg.1: "... While recent LLM-based agents have demonstrated remarkable success in autonomous experiment planning, a robust workflow for the faithful digitization and verification of existing literature remains a challenge. Our approach bridges this gap by providing ..."

pg.2: "... , notably, recent work has demonstrated the power of agents to autonomously plan syntheses via web search¹⁷ or orchestrate expert software tools to solve complex chemical tasks¹³ ...",

pg.3: "... While early agents^{13,17} have excelled at *de novo* experiment planning and direct code generation for specific platforms, a critical challenge remains in the faithful digitization of the existing literature. Unlike prior efforts that focus on generating new routes or utilizing platform-specific APIs, our workflow is designed to standardize the vast corpus of unstructured reported procedures. ACRA enables the transition all the way from a static literature document to the execution of a synthetic procedure by translating into the hardware-agnostic XDL standard, and importantly, introduces a simulation layer to validate that the generated instructions match physical hardware constraints. ...",

Q2: The description of the automated experimental platforms is overly brief, and the Supplementary Information also lacks technical details. The authors should elaborate on how automated syntheses were executed in practice to improve reproducibility and transparency.

We provided more details in the SI Section 5 on implementation of the automated chemistry platforms. The notebooks and log files automatically generated during execution are now also available as part of the Zenodo repository and shared files for reviewers.

Q3: Although several successful demonstrations are presented, many literature procedures could not be reproduced, especially those requiring TLC steps, which frequently appear in the failed cases. The authors should provide a systematic analysis of these unexecuted steps and failure modes, identifying underlying causes and suggesting possible solutions.

During our benchmarking we have identified several steps that are common, yet not executable in our Chemputer platform. This is addressed in the manuscript section “Systematically improving XDL by identifying non-executable synthetic steps“ and visualised in Figure 6. We believe that this concern is already addressed in that section as we identify through survey most common steps missing implementation in XDL, as for example here mentioned TLC analysis. Importantly, the fact that the chemputer platform cannot execute some step does not undermine the pipeline, it is primarily an engineering problem/challenge and motivation to implement that to broaden portfolio and make the robots even more general.

Q4: The Critique-Agent employs an LLM-as-a-judge approach. Are there instances of misjudgment or oversight in its decisions?

Critically, the safety of execution is captured by the simulation, not judge. Importantly, we show the overall translation accuracy improves when judge is used (see Figure 4E). We believe the use of judge is thus tested/benchmarked.

Are there user confirmation or rollback mechanisms in place?

User confirmation is required for the start of the synthesis, at which stage intervention and/or procedural edits may be undertaken.

Q5: The overall performance of the “literature-to-experiment” pipeline could be more clearly quantified. The authors are encouraged to report success rates at each stage to demonstrate the specific contribution of each agent and the collaborative workflow.

End-to-end translation from paper to executable and tested XDL was benchmarked in Figure 4D/F directly from primary literature already provided procedures, and without the LLM-as-a-judge as an ablation study. All other parts (substeps) were benchmarked:

- Procedure extraction was tested in Figure 3C, and classification in Figure 3B and S23/24.
- Knowledge-graph extraction (chemical entities & analytical data) in Figure 3E/F.
- Efficiency in number of iterations to respective XDL translation stage in SI section 1.3.2-1.3.4
- The addition of external data sources was ablated in Figure 5.
- Language depended sanitization/translation capabilities were tested in Figure S25.

We believe that this showcases the performance of all part of the collaborative workflow.

Response to Reviewer 2

1. While the authors have clearly put a lot of thought and effort into this manuscript, I believe the work should be framed as a (very thorough) proof of concept. Stating this more clearly in the manuscript would help set the right expectations for the reader. I suggest adding a sentence in these regards in either the abstract, introduction, or conclusion.

We thank the reviewer for the suggestion and adapted the conclusions to clarify the purpose of our work (Conclusion pg. 19 "... While this work serves as a thorough proof of concept showcasing the potential for integrating LLMs into autonomous chemical synthesis, we hypothesize that such frameworks can be expanded to incorporate more simulation data, eventually enabling LLM-based agents to orchestrate entire laboratories. ...")

2. The authors present their findings in a positive light, and I agree that their results are very encouraging. At the same time, it would be interesting to know more about the challenges and what isn't working so well yet. For example, the authors could discuss the system's current readiness for multi-day or multi-step syntheses and give readers a better idea of the future improvements they envision (beyond just extending to non-textual data).

We extended the discussion to address remaining challenges we faced during our work, hindering fully autonomous scientific reproduction (pg. 20 "... However, it is important to highlight, that challenges in fully autonomous execution of scientific literature still face experimental challenges such as automated purification of unknown mixtures, automated analytics, or even reagent availability, though some solutions have been proposed¹⁻³. Robust solutions to these challenges allow to extend LLM-controlled autonomous systems to execute many-step, successive synthesis procedures. ...")

3. The authors should discuss the classification accuracy of 67.5% in more detail, as this seems to me to be a major bottleneck. In my view, the manuscript doesn't fully address the implications of this performance level. I also think the authors should better describe how a misclassification (e.g., an 'executable' procedure being flagged as 'incomplete') is handled downstream in the pipeline.

We added a confusion matrix (Fig. S24) showing the translation across classes (actual vs predicted). As becomes clear from that, most misclassifications stem from the classification of "blueprint procedures" as "complete procedures". We additionally extended our manuscript discussing this shortcoming ("... 75 % of failure cases resulted from ambiguously referenced chemicals or references to general procedural instructions. Thus, cases of misclassification stem from *blueprint procedures* being classified as *executable procedures* (9 out of 12; SI Fig S24). These procedures will be translated into XDL, but missing essential chemical entities hindering physical execution. ...")

4. Along the lines of performance metrics, I would be curious to know if the authors considered comparing their extraction performance with other published tools (ChemDataExtractor, which is able to handle multimodal input to some extent). They should provide a reason why a comparison to one of these tools wasn't included here.

We thank the reviewer for this valuable suggestion. To address this, we have included a benchmark comparing ACRA against ChemDataExtractor 2.0 (CDE) on the shared task of Chemical Entity Recognition (see Figure 3E), where ACRA demonstrates highly competitive performance. While CDE excels at extracting static chemical properties (e.g., melting points) for database construction etc., ACRA is specifically optimized to extract procedural logic and associated data (such as chemical entities) in a narrower domain. Additionally, we highlight that the physical execution of reactions serves as a critical, functional validation for autonomous agents, complementing standard text-mining metrics.

5. The "Chemical Ambiguity Database" (CAD) is a great addition for providing "expert

chemical intuition." The study mentions it is initialized based on just 5 annotated procedures, which is separate from the XDL Database used for few-shot examples. To me, this raises a question about the scalability of this process. I think the authors could add a brief discussion about this, considering whether it might be desirable to have a very large ambiguity database in the future, or if the current approach is a good enough trade-off to keep the workflow lightweight.

Done, we extended our discussion with a brief comment on the scalability of the CAD to allow the reader to understand the limitations (pg20 "... Although the CAD was valuable (compare Fig. 5), its need for extensive manual curation limits practical expansion, even though the current data already resolves the most common pitfalls we initially observed. ...")

6. I think the authors should elaborate more on the "autocorrection" feature, such as changing 30 extractions to 3 in Figure 7D. This is a powerful but potentially risky capability. It would be interesting to discuss how the system distinguishes a reporting error from a genuine (though unusual) procedure and what safeguards are envisioned to prevent incorrect "fixes".

We agree that this "autocorrection" while having potential, poses the risk of falsely changing unusual, but correct procedures. This is however, less of a feature, but more the inherent nature of the underlying LLM. Since the system is provided with the most similar previously provided procedures, we envision that this problem will be largely solved through a large enough sample of validated procedures. In case a "fix" leads to irreproducibility, the procedure would remain in the set of unvalidated procedures to be revisited or corrected later. We adjusted the wording of the respective section to highlight this ambiguity ("... To verify that the missing conversion was not the result of a false correction, the synthesis was additionally executed with varying hardware implementations by a synthetic chemist closely following the reported procedure. ...")

7. Along the same lines, the workflow makes quantitative approximations for ambiguous phrases (e.g., "dropwise transfer" becomes "10 min" or "lifted half way" becomes a 15°C temperature drop). The authors should clarify how these specific values are reasoned.

We expanded our manuscript, commenting on the quantitative approximations (pg18 "... As evident from the validated examples, the qualitative nature of reported language necessitates a "best guess" approximation to define executable parameters (e.g. "dropwise transfer" to a 10-minute duration). Since this is an inherent problem of how chemical procedures are reported, we conclude that the provided experimental validation suggests that these system-derived approximations are chemically reasonable and effectively resolve the inherent ambiguities of the source text in many cases. ...")

8. The authors could reframe the failed synthesis in Figure 7D. In my view, the failure to reproduce this reaction is a success for a verification-focused workflow. It's a demonstration of ACRA's utility in identifying non-reproducible procedures from the literature, and a (cautious) statement in this direction could be added in my opinion.

We thank the reviewer for positive interpretation of our validation pipeline. The manuscript was adapted to properly represent this demonstration (pg18 "... Nevertheless, the reported reaction could not be verified without substantial alterations to the procedure and was thus classified as not reproducible highlighting the potential of the presented system. ...")

9. The authors mention in the SI (Section 2.6) that open-source models failed to generate valid XDLs. This is an important finding for the readership. I suggest the authors briefly move this result into the main manuscript's discussion, as it highlights the workflow's current dependence on proprietary models.

We expanded our manuscript to discuss this important finding (pg8 "... We additionally tested two frontier open-source language models with up to 70b parameters as the model behind the XDL agent. Neither of the models generated valid XDL however (compare SI section 2.6 for an output comparison). ...")

Response to Reviewer 3

Unclear Optimization and Benchmarking

The article does not adequately explain how various parameters (e.g., chunk size, data splitting) were optimized.

In the case of chunk sizes, we have tested several values, and we have a discussion why we have chosen the specific chunk sizes in the SI (pg. 5-6). We have made conservative choice about chunk size as well as overlaps. This is in line with current know how (<https://learn.microsoft.com/en-us/azure/search/vector-search-how-to-chunk-documents>).

For instance, it reports benchmark data but fails to specify if these datasets are representative, optimized, or simply test data from previous work. This raises concerns that the results may not be reproducible or rigorously tested.

We provide a discussion of the dataset for each benchmark in the SI. The knowledge graph extraction (Figure 3E) as well as the translation accuracy (Figure 4F) the dataset presented in Mehr *et. al.*⁴ was used as mentioned in the manuscript, as well as the SI. The translation efficiency (Figure 4D), as well as the ablation studies (Figure 5B) were performed with randomly sampled procedures from the ChemRnD dataset (compare manuscript and SI). The procedure classification (SI Figure S23), as well as extraction of analytical data (Figure 3F) from randomly sampled procedures from the set of procedures in the procedure extraction to cover a diverse range of sources.

Figure captions showing benchmark results should specify the support set size and how error bars were obtained.

We added respective description to all relevant figure captions (main manuscript: Figures 3,4, and 5. SI Figures S19, S20, S21)

Biased and Incomplete Datasets

The evaluation datasets appear highly biased, with only eight procedures used for the knowledge graph extraction part. No detailed scoring or criteria are provided for how tolerances were set (e.g., when temperatures are deemed equivalent), leading to potential inaccuracies in the extracted data. This may result in skewed performance metrics that don't reflect real-world applicability.

For knowledge graph extraction - we have examples of tolerance e.g. for analytical data split J coupling NMR in the SI, page S35-36. For extraction of analytical data, we chose random

sample of 20 procedures (from the set of procedures discussed in Figure 3B and Figure 3C in the manuscript). For translation to XDL, and extraction of chemical entities, the procedures and methodology of translation accuracy is adapted from our previous Science paper⁴. We additionally extended this analysis with a comparison to ChemDataExtractor2.0 (Figure 3E).

Issues with Supplied Data/Code

The supplied Dropbox is extensive but poorly organized, lacking a proper README file or clear instructions. This makes it challenging to navigate the data, as some folders are empty, and log files contain inconsistencies (e.g., a product mentioned that isn't in the extraction). The absence of detailed scores for the knowledge graph part further complicates assessment of the method's effectiveness.

Thank you for checking in on the details; we appreciate the feedback and have addressed the concerns as described below.

We added substantial amount of additional description of the provided code with extended README files for the code, explaining the expected output, and available notebooks (##Experiments section in code/README.md). We additionally added a README file to the root-folder explaining the provided data and locations, and subfolders where applicable. We additionally expanded the provided computational data, as well as synthesis related data.